# Impact of Long-Term Care Service Quality on User Cooperation/Participation: Mediating Effect of Self-Determination

**DOI:** 10.3390/healthcare12020193

**Published:** 2024-01-13

**Authors:** Hanra Cho, Junsu Kim

**Affiliations:** Department of Social Work, Seowon University, Cheongju 28674, Republic of Korea; halla_c@daum.net

**Keywords:** community care, long-term care, self-determination theory, service quality, participation, cooperation

## Abstract

The purpose of this study is to determine whether home care service quality affects users’ voluntary cooperation/participation through self-determination. For this purpose, survey data from 358 long-term care service users in Korea were analyzed by applying a structural equation model. Data collection was conducted from February to March 2019. Data collection was conducted in February–March 2019. The results showed that home care quality did not directly affect users’ voluntary cooperation/participation in the service delivery process, and self-determination fully mediated the relationship between quality and cooperation/participation. Based on these results, the importance of providing appropriate information, reflecting users’ voices, and the attitude and role of experts was emphasized in order to promote user participation/cooperation through self-determination.

## 1. Introduction

Korea has entered an aging society, with 14.3% of its population aged 65 and over in 2018, and this is expected to reach 18.4% by 2023 [1]. Korea’s aging population is aging at an unprecedented rate. Older adults prefer to age in place, requiring a high level of community support [2]. In response, the Korean government is focusing on building community care [3], and the importance of the long-term care insurance system for the elderly, which has been in place since 2008, is expected to grow.

Korea’s long-term care system operates based on the principle of social insurance. This is different from the care system that was previously selectively provided to low-income older adults in the form of public assistance. Specifically, in terms of service users, services that were previously only provided to older adults below a certain income standard can now be used by anyone, regardless of income. In the past, the system was funded solely by taxes, but the long-term care insurance system is funded by social insurance and a certain amount of co-payment. As for the delivery system, care services provided by public assistance were mainly handled by non-profit corporations, but now individuals can also run organizations. Services have also expanded to include various services, such as home care, home bathing, and home nursing. This can be said to be a similar trend to the social welfare service delivery model, which developed from a professionalism model to a consumerism model based on economics, giving users a choice [4,5]. This means that users can choose which services they want to receive and from whom, based on their needs, as opposed to having a professional determine what services they need through an assessment. The aforementioned changes in coverage, co-payments, and reduced barriers to entry for service providers will play a major role in this transformation.

These services are available to anyone who has a need for care due to geriatric diseases, etc., and emphasizes the user’s right to choose by applying market principles. In other words, this is a similar trend to the evolution of the social welfare service delivery model from a professionalism model to a consumerism model based on an economic perspective, giving users the right to choose [4,5].

However, the commoditization of services and the expansion of options alone cannot be considered to provide services that truly reflect the interests of users [6]. Therefore, the co-production model, in which experts and users share the decision-making process, is gaining attention [7,8]. Especially in cases where long-term support is needed, needs are constantly changing, or services have a major impact on the user’s quality of life, the co-production model is more appropriate than the traditional provider-centric or consumer model [9]. This co-production model emphasizes service user participation and cooperation for several reasons. First, because service users are the most knowledgeable about their needs [10], utilizing them as collaborators helps to improve services to ensure that the goals of the service are well met [11]. Second, because home care services are provided in the user’s home and are tailored to their individual needs, interaction is inevitable [12], and that interaction determines the amount and nature of the care and the work of service providers [13]. In other words, client cooperation facilitates successful interactions with service providers at the service encounter [14]. Third, participation encompasses both the user’s suggestions, ideas, complaints, suggestions, etc., for the development of the organization’s policies, which makes the service organization more aware of the problems they have so that they can create better services [15].

So, what makes users cooperate and engage in the service process? Previous research has shown that service quality influences users’ willingness to participate and cooperate [11,16]. Other studies have reported that quality affects self-determination [17], and that self-determination at the service contact point affects service users’ cooperation and participation [14,18]. In this context, self-determination is a widely used concept in the field of motivation theory, referring to the ability to make choices and determine actions on one’s own, without being coerced by rewards or external pressures [19,20].

Self-determination theory argues that motivation is made possible by understanding our innate psychological needs [21]. Self-determination theory is composed of several mini-theories, of which the basic psychological needs theory consists of autonomy, competence, and relatedness. Autonomy is the most important concept and is the belief that one is the subject and regulator of one’s own actions [18]. Competence is more focused on the psychological motivation aspect and is the belief that one can interact effectively with the environment [20], a similar concept to self-efficacy, as proposed by Bandura (1986) [22]. Finally, in addition to being competent and free, humans also seek to feel meaningfully connected to others in a social context [20,23], which is the relatedness need.

This relational need is even more important in care relationships [24], because care is not just about physical care, but also about caring for psychological, emotional, and developmental needs [25], and care providers and recipients become interdependent and relational beings [26]. In other words, a strong relationship between service providers and users facilitates the exchange of information about each other, and helps them to identify mutual needs and discover opportunities to create new values based on trust in each other [27]. This, in turn, is conducive to user participation and cooperation.

The self-determination theory has been used to understand the psychological motivations of customers or service users in a variety of fields, including education [28,29,30], service marketing [17,18,31,32], and social services [33,34,35]. In the context of older adults, self-determination theory has been applied in a variety of fields, including research on aging, adaptation, and leisure among community-dwelling older adults [36], research on the effectiveness of nursing home programs [32], and the motivation to use mobile health applications [37]. However, there is a lack of research on the care of older adults living in the community. In addition, studies on user participation and cooperation in social services have mainly focused on theoretical discussions [7,8,10,38] and have not empirically confirmed whether self-determination affects user participation and cooperation. Therefore, this study aims to examine whether self-determination mediates the relationship between service quality and participation and cooperation, based on basic psychological needs theory.

## 2. Materials and Methods

### 2.1. Research Model

A research model was set up, as shown in Figure 1. The research questions of this study are as follows.

First, does the quality of home care service affect users’ participation/cooperation?

Second, does users’ self-determination mediate between quality and participation/cooperation?

### 2.2. Research Subjects and Data Collection

The target population of this study is people aged 65 and over, who use home care services under long-term care insurance. The sampling method was based on the list of organizations disclosed on the website of the long-term care insurance, and the distribution of organizations in 14 cities and counties in Jeollabuk-do was considered and allocated by region. To ensure the ethical protection of research participants, data were collected with the approval of the institutional review board. The data collection period was February–March 2019. Data were collected face-to-face after receiving written consent to participate in the study. Participants filled out a structured questionnaire or answered questions with the help of family members and others. A total of 457 copies of the questionnaire were distributed and a total of 359 data were collected. Of the collected data, 358 data were used in the final analysis, with the exception of one outlier for some variables.

### 2.3. Definition and Measurement of Variables

#### 2.3.1. Cooperation/Participation

Several scholars describe the concept of participation as the degree to which users are involved in the service delivery process [39] or as a behavior [40], as a temporary member with a certain role and as a co-creator [41]. In this study, cooperation and participation refers to whether the user voluntarily cooperated and participated, in addition to the required behavior in the process of using the service, and the items used in Bettencourt’s (1997) study [15] were modified and supplemented for this study.

Cooperation was measured by three questions about “helping the staff, following the rules and policies of the organization, and treating the staff with respect and kindness”. Participation was measured by three questions about “giving advice on how to help the organization better understand the needs of the users, giving useful ideas to improve the service, and praising the staff when they did a good job”. The response categories ranged from “not at all” to “very much”. 

Responses ranged from 1 (not at all) to 5 (very much so), with higher scores indicating higher levels of engagement and cooperation. The reliability of the total six items of the cooperation and participation scale was secured with Cronbach’s α = 0.765.

#### 2.3.2. Self-Determination

Self-determination utilizes a scale developed by Deci and Ryan (2000) to measure basic human psychological needs, applicable to both relationships and life in general [21]. The scale consists of autonomy, competence, and relatedness (http://www.psych.rochester.edu/SDT/measures/needs_scl.html, accessed on 30 October 2023). In this study, it was modified and supplemented by the relationship between home care service users and service providers (social workers and caregivers).

Autonomy was measured with 7 items, including “I feel free to decide how to use the service”, “I use the service in my own way”, and “I don’t have many opportunities to decide the service in my own way (reverse)”. Competence was measured by 6 items, including “I am confident in my ability to use the service” and “Others recognize that I am competent in using the service. Relationship was measured by 6 questions about the relationship with the social worker and 6 questions about the relationship with the caregiver, such as “I get along well with my social worker (caregiver)”, “I get along with my social worker (caregiver)”, “I get along with my social worker (caregiver)”, and “I get along with my social worker (caregiver)”. The response categories ranged from “not at all” to “harmonious”. 

The responses were scored on a Likert scale ranging from 1 (not at all) to 5 (very much so). The higher the score, the higher the self-determination, and the reverse questions were recorded and analyzed. The reliability of the 25 items of the self-determination scale was high, with Cronbach’s α = 0.843.

#### 2.3.3. Service Quality

Service quality refers to the user’s subjective evaluation of the service and was measured using Cronin and Taylor’s (1992) SERVPERF model [42]. In the SERVPERF model, quality is composed of five dimensions, including tangibles, which are related to the physical conditions of the organization; reliability, which is the ability to perform the service accurately; responsiveness, which is the ability to help quickly and actively; assurance, which is the ability to give trust and confidence in the service, such as the knowledge and courteous manners of the staff; and empathy, which is the care and attention that the organization provides to the user, and good communication [42]. However, this study did not measure tangibility, which is a quality measure of an organization’s facilities and equipment, given that the place where services are provided is the user’s home. 

A total of 16 questions were used to measure reliability, consisting of 5 questions on reliability, 3 questions on responsiveness, 4 questions on certainty, and 4 questions on empathy. The response categories were scored on a Likert scale ranging from 1 (not at all) to 5 (very much so). Higher scores indicate higher quality. The reliability of the 16-item quality scale was found to be very high, with a Cronbach’s α = 0.956.

### 2.4. Methods

The data were analyzed using IBM’s SPSS 25.0 and AMOS 25.0 programs. First, the data were checked for outliers and missing values and checked for normality, and the characteristics of the research subjects and the main variables were checked through frequency analysis and descriptive statistics. After confirming whether the research model was suitable for applying the structural model through a measurement model, the structural model was verified using a structural equation model.

## 3. Result

### 3.1. Characteristics of Research Subjects and Main Variables

The general characteristics of the subjects and the characteristics of the main variables are shown in Table 1. Of the 358 participants, 264 (73.7%) were female and 94 (26.3%) were male. The majority of the respondents were in their 80s, 178 (49.7%), and the average age was 81.51 years old. In terms of education, 151 (42.2%) had no education, 111 (31.0%) had a high school diploma, 54 (15.1%) had a high school diploma or higher, and 42 (11.7%) had a college degree. In terms of household type, 169 (47.2%) were living alone, and the average weekly usage time was 12.6 h.

For service quality, the highest score was 4.31, for certainty. Among the self-determination dimensions, relationship was the highest, at 4.10, followed by autonomy at 3.52, and competence at 3.50. Cooperation was found to be 3.76, and the participation score was 3.39. The skewness of the main variables ranged from −0.977 to 0.009 and the kurtosis ranged from −0.397 to 3.302, indicating that the data were normal according to Kline’s (2005) criteria [43].

### 3.2. Measurement Model

The goodness of fit of the measurement model was χ^2^ = 74.410 (df = 24, *p* = 0.000), but the χ^2^ statistic is sensitive to the sample size and overestimates the discrepancy between the model and the data [44]. Therefore, we further checked the model fit, which is not sensitive to the sample size. An RMR (RMSR) of 0.05 or less, an RMSEA of 0.08 or less, and a GFI of 0.9 or more are considered a good model fit [45]. In this study, the absolute fit index of the measurement model was found to be RMR = 0.015 (RMSR = 0.042), RMSEA = 0.074, and GFI = 0.957.

Next, to verify the construct validity of the variables, we examined the convergent validity, which refers to the agreement of the observed variables measuring the latent variable, and discriminant validity, which refers to the difference between the different latent variables. The standardized factor loadings of service quality ranged from 0.708 to 0.902, with AVE = 0.881 and concept reliability = 0.967. In addition, the standardized factor loadings of self-determination ranged from 0.492 to 0.747, and although the standardized factor loadings of autonomy did not meet the criterion, the AVE = 0.697 and conceptual reliability = 0.870 indicated that the focus validity was secured. The standardized factor loadings of cooperation and participation ranged from 0.650 to 0.860, with AVE = 0.725 and conceptual reliability = 0.838, indicating that all variables included in the study were convergent. The results of the measurement model are presented in Table 2.

As for discriminant validity, we compared the AVE and ∅^2^ values between all variables, as shown in Table 3, and found that the AVE value was greater than the ∅^2^ value, and [∅ ± 2 × S.E.] did not contain 1. Therefore, the discriminant validity of the input latent variables was secured.

### 3.3. Structural Model

The results of the structural model analysis are as follows. The fit of the structural model was χ^2^ = 74.410 (df = 24, *p* = 0.000), RMR = 0.015 (RMSR = 0.042), RMSEA = 0.074, GFI = 0.957, NFI = 0.953, IFI = 0.968, TLI = 0.952, CFI = 0.968. The path coefficients and significance results of each path, as identified in the structural model analysis, are shown in Table 4.

Specifically, the path from quality to self-determination was positively related and statistically significant (β = 0.827, *p* = 0.000), meaning that the higher the quality of the home care service, the higher the self-determination. The path from self-determination to cooperation and participation also showed a positive relationship and was statistically significant (β = 0.485, *p* = 0.006). This means that the higher the level of self-determination among users, the higher the level of cooperation and participation. However, the path from quality to cooperation and engagement is not statistically significant. Therefore, we can say that self-determination fully mediates the relationship between quality and cooperation and participation, and that good quality alone does not lead to users’ voluntary cooperation and participation.

The results of the statistical significance of the mediating effects using the bootstrapping method are shown in Table 5. The total effect of the path from quality to participation and cooperation was 0.424, the direct effect was 0.023, and the indirect effect was 0.401. The indirect effect is significant at the 0.05 level, as the bootstrapping bias corrected (BC) is not zero between the lower and upper bounds (0.47~1.120), with a 95% confidence interval [45]. Therefore, the indirect effect of self-determination on the path between quality and participation and cooperation is statistically significant.

## 4. Discussion

The purpose of this study is to determine whether home care service quality affects users’ voluntary cooperation and participation through self-determination. For this purpose, data from 358 long-term care service users in Korea were analyzed using the structural equation model.

The main findings of this study are as follows. First, the quality of home care services does not directly affect users’ voluntary cooperation and participation in the service delivery process. This means that the quality of services provided by caregivers or social workers alone is not sufficient to elicit active cooperation and participation from users.

Second, quality was found to affect self-determination. This is consistent with the results of a study examining the relationship between service quality and self-determination using data on daycare center users, which showed that service quality affects self-determination behaviour [46]. It is also consistent with a study that examined the relationship between service quality and satisfaction mediated by self-determination in a model of mobile banking users, and found that service quality influences self-determination [17].

Third, self-determination was found to influence cooperation and participation, i.e., self-determination, which consists of users’ autonomy, relatedness, and competence, was found to increase cooperation and participation at service encounters. These findings support Morgan and Hunt’s (1994) study, which argued that relationship development variables such as commitment and trust can explain service user cooperation [27], and are consistent with a study that found that hospital service users’ self-determination (autonomy, competence, and relatedness) influenced customer engagement [18].

Fourth, we find that self-determination mediates the relationship between quality and cooperation and participation. In other words, the direct path from quality to cooperation and participation was not statistically significant, and self-determination fully mediated the relationship between quality and cooperation and participation. This is similar to existing research results showing that service quality does not directly affect users’ voluntary actions, such as participation and cooperation, and that participation and cooperation require a medium that triggers voluntary actions such as satisfaction and trust with the service [11,16]. In other words, while providing good quality services is a prerequisite, good quality alone does not lead to voluntary cooperation and participation. User participation and cooperation can be implemented only when users have the mindset that they can choose and use services according to their own will and maintain a good relationship with the service provider.

The significance and implications of this study are as follows. First, we found that self-determination is an important variable that can explain the process from quality to cooperation and participation in a more concrete way. Until now, the discussion of user participation and cooperation in the care sector has only been theoretical. However, the significance of this study is that it empirically verifies the factors that lead to user participation and cooperation. Self-determination theory has mostly been applied in the fields of education [28,29,30], service marketing [18,31,32], and social services [33,34,35], but this study validates the utility of self-determination theory in the field of community-based older adult care.

Therefore, service providers should provide appropriate information to increase users’ autonomy and show support and encouragement for their choices. It is more effective to present a few options that are suitable for the person than to present too many options. When selecting options, the social worker or caregiver should prioritize the interests of the person, rather than the interests of the organization or the convenience of the service delivery. Given the importance of interaction in enhancing self-determination [21], efforts should be made to ensure that the service user–provider relationship is supportive.

In addition, users must be involved in all stages of service assessment, planning, monitoring, and evaluation, and their opinions must be actively reflected so that they can experience competence. Of course, currently, users’ participation and cooperation are encouraged in the above stages, but user participation in long-term care services is still at the stage of nominal formal participation or users being informed about services. In other words, services are still planned and led by professionals such as social workers. In order to promote user participation, the attitudes and roles of professionals, such as recognition and encouragement of participation, along with institutional arrangements that ensure user participation activities, are important [47]. Therefore, it is necessary to establish a structure for direct user participation at all stages of service assessment, planning, monitoring, and evaluation, and service providers should have confidence in the competence of users and support them in making decisions on an equal footing.

Finally, this study is limited in generalizability because the sample selection was limited to home care service users in Jeollabuk-do, Korea. In addition, the quantitative approach did not allow us to explore a wider range of variables affecting service user cooperation and engagement, nor did it allow us to explore participants’ perspectives in more depth. We hope that future research will address these limitations.

## 5. Conclusions

In order to promote user participation and cooperation through service quality and self-determination in long-term care services, a representative older adult care system in Korea, appropriate information should be provided to increase user autonomy. In addition, users’ voices should be actively reflected so that they can experience competence, and the attitudes and roles of professionals such as recognizing and encouraging users’ participation and cooperation are important.

## Figures and Tables

**Figure 1 healthcare-12-00193-f001:**
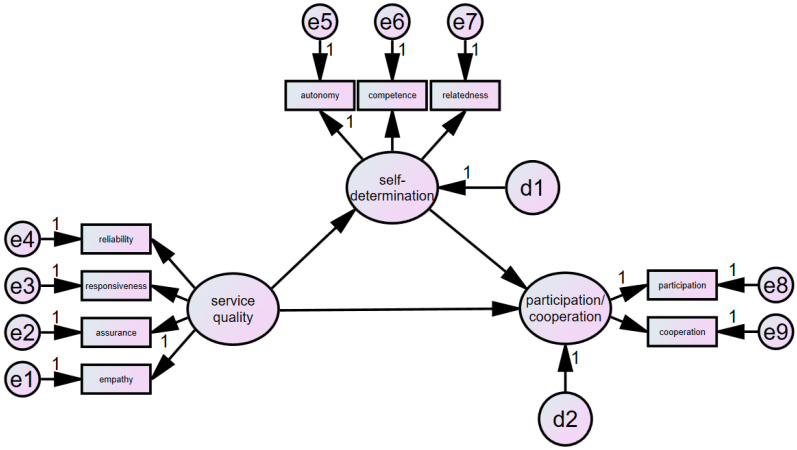
Research model.

**Table 1 healthcare-12-00193-t001:** Characteristics of research subjects and main variables.

Variable		N	%	Variable		M	S.D.
Gender	Male	94	26.3	Age		81.51	8.22
	Female	264	73.7	Average Weekly Usage Time		12.6	4.29
Age Range	60s	30	8.4	Service Quality	Reliability	4.22	0.55
	70s	106	29.6		Responsiveness	4.28	0.54
	80s	178	49.7		Assurance	4.31	0.53
	Over 90s	44	12.3		Empathy	4.15	0.55
Education	No Edu.	151	42.2	Self-Determination	Relatedness	4.10	0.61
	Elementary	111	31.0		Autonomy	3.52	0.49
	Middle	42	11.7		Competence	3.50	0.51
	High or Higher	54	15.1	Participation/Cooperation	Participation	3.39	0.76
Form of Cohabitation	Living Alone	169	47.2				
	Living Together	189	52.8		Cooperation	3.76	0.63

**Table 2 healthcare-12-00193-t002:** Convergent validity of observed variables.

Latent Variable	Measurement Variable	Estimate		S.E.	CR	AVE	Concept Reliability
		B	*β*				
Service Quality	Reliability	0.890	0.708	0.048	18.695 ***	0.881	0.967
	Responsiveness	0.961	0.852	0.044	21.872 ***		
	Assurance	1.000	0.902				
	Empathy	0.924	0.804	0.047	19.694 ***		
Self-Determination	Relatedness	1.000	0.747			0.697	0.870
	Autonomy	0.524	0.492	0.064	8.194 ***		
	Competence	0.691	0.621	0.068	10.183 ***		
Participation/Cooperation	Participation	0.906	0.650	0.138	6.544 **	0.725	0.838
	Cooperation	1.000	0.860				

Model fit: CMIIN = 74.410, DF = 24, CMIIN/DF = 2.975 (*p* = 0.000), RMR = 0.015 (RMSR = 0.0426), RMSEA = 0.074, GFI = 957, NFI = 0.953, IFI = 0.968, TLI = 0.952, CFI = 0.968. *** *p* < 0.001, ** *p* < 0.01.

**Table 3 healthcare-12-00193-t003:** Discriminant validity of observed variables.

Variable	Correlation Coefficient	Comparison of AVE and ∅^2^		∅ ± 2 × S.E.
Service Quality ↔ Self-Determination	0.827	Service Quality AVE = 0.881	>0.684	0.787~0.867
		Self-Determination AVE = 0.697		
Self-determination ↔ Participation/Cooperation	0.503	Self-Determination AVE = 0.697	>0.253	0.467~0.539
		Participation/Cooperation AVE = 0.725		
Service Quality ↔ Participation/Cooperation	0.424	Service Quality AVE = 0.881	>0.180	0.384~0.464
		Participation/Cooperation AVE = 0.725		

**Table 4 healthcare-12-00193-t004:** Structural model analysis results.

Variable Path	Estimate		S.E.	C.R.
	B	*β*		
Service Quality → Self-Determination	0.792	0.827	0.060	13.249 ***
Self-Determination → Participation/Cooperation	0.578	0.485	0.210	2.747 **
Service Quality → Participation/Cooperation	0.026	0.023	0.182	0.142

Model fit: CMIIN = 74.410, DF = 24, CMIIN/DF = 2.975 (*p* = 0.000), RMR = 0.015 (RMSR = 0.042), RMSEA = 0.074, GFI = 957, NFI = 0.953, IFI = 0.968, TLI = 0.952, CFI = 0.968. *** *p* < 0.001, ** *p* < 0.01.

**Table 5 healthcare-12-00193-t005:** Effect decomposition.

Variable Path	Total Effect	Direct Effect	Indirect Effect	Confidence Interval
Service Quality → Participation/Cooperation	0.424	0.023	0.401	0.14~1.120 *

* *p* < 0.05.

## Data Availability

The authors own this data and agree to make the materials and data available to the journal and other researchers upon request.

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
