# Peer review of "Impact of Long-Term Care Service Quality on User Cooperation/Participation: Mediating Effect of Self-Determination"

_healthcare, 2024, doi:10.3390/healthcare12020193_

Round 1

Reviewer 1 Report

Comments and Suggestions for Authors

This paper underscores the role of self-determination in the context of older adults utilizing long-term care services, a topic of growing importance due to demographic changes. Its application of self-determination theory to examine participation and cooperation in this realm yields valuable insights.

However, the paper could be enhanced by:

  1. Expanding the introduction to detail the development of Korea's long-term care insurance system. clarifying users' rights in choosing either the insurance plan or the service provider and (line 30-32).

  2.  

  3. Addressing the potential influence of demographic factors (age, education, cohabitation) shown in Table 1 on self-determination and participation/cooperation, and whether these were considered in the mediation analysis.

  4.  
  5. Resolving inconsistencies in table numbering, notably the absence of Table 5 and the unexplained reference to Table 6 (Line 223).

Comments on the Quality of English Language

The paper generally maintains a clear and coherent structure in writing. However, some sections could be clearer and more concise. For example, in Lines 224-227, "The significance of the indirect effect is significant at the p<.05 level", can be shortened as "The indirect effect is significant at the 0.05 level". It is also not clear to me whether the 95% CI (0.47-1.12) is for the indirect effect coefficient.

Author Response

Thank you for your review. 

Reviewer 2 Report

Comments and Suggestions for Authors

Thank you for the opportunity to review this manuscript on a relevant topic related to older care service quality. I would like to leave my suggestions and questions.

Abstract

Please, enter the date of study in the abstract.

Material and Methods

-Lines 96-97: “The subjects of the study are users of home care services under the long-term care insurance for the elderly.”  As an inclusion criterion, the authors considered the elderly from what age? Please describe the age.

-Lines 102-103: “After receiving written consent to participate in the study, participants filled out a structured questionnaire or answered questions with the help of family members and others.”  Was the questionnaire applied face-to-face or online? If online, where was the questionnaire available? Please describe.

-Lines 104-105: A total of 359 data were collected and, excluding one outlier, a total of 358 data were used for the final analysis.” From how many invited participants, the authors obtained 359 participations? Please describe the total number of participants who were invited to participate in the study.

-Lines 118-121: “The response categories were "not at all" to "very much. Responses ranged from 1 (not at all) to 5 (very much so), with higher scores indicating higher levels of engagement and cooperation."  Was there neutral answer (do not know/ do not answer)?

-Lines 141-142: “The reliability of the 17 items of the self-determination scale was 141 high with Cronbach's α=.843.”  Were there 17 items or 25 items (7 items about autonomy + 6 items about competence + 6 items about relationship with the social worker + 6 items about the relationship with the caregiver)?

-I suggest the availability of the questionnaire with all items included. This questionnaire may be included as an Appendix.

Results

-Line 177: “Cooperation was found at 3.76 and participation at 3.39.”  This information is shown in Table 1 in reverse. Please rectify.

-Lines 196-197: “The standardized factor loadings of cooperation and participation ranged from .650 to .865,…” This information shown in Table 2 is PARTICIPATION as .650 and COOPERATION as .860 (not .865). Please rectify.

-Line 218: “Therefore, we can conclude that self-determination fully mediates the relationship between quality and cooperation…” Conclude or verify? Is it right write “conclude” in Results section?

-Lines 222-223: “The results of the statistical significance of the mediating effects using the bootstrapping method are shown in Table 6.” Where is Table 6?

-Tables could be presented after their in-text citation.

 Discussion

Another limitation of the study is the quantitative approach, which does not allow for a deeper exploration of the participants' perspectives.

Author Response

Thank you so much for your thorough review.

Reviewer 3 Report

Comments and Suggestions for Authors

Thank you very much for the opportunity to review this article.

The article reflects interesting work and may be of interest for future developments. One crucial aspect is the authors' ability to synthesize, which allows for a much more straightforward reading of the text, condensing their findings quite well. However, I think several minor aspects could be revised to improve the readability and overall quality of the article:

The authors should structure the abstract better, clearly separating the different sections. For example, I would suggest they remove the phrase "The main results are as follows," as it does not provide information and separate the part intended to present the conclusions. I would also invite them to show the main numerical results obtained so the reader can understand them from the beginning.

In the methods section, the authors can be more systematic, going from the most general to the most specific aspects of the study so that readers can perfectly contextualize the methodology, which is essential to evaluating the potential risks. First, the study type conducted should be defined, and then the eligible population, inclusion and exclusion criteria, etc., should be shown.

In point 2.1. the authors describe a research model but do not provide information about its development. This information, even summarized, would add significant value to their methodology since it must be reproducible by other researchers.

Still, in methods, it is essential to describe in greater detail the eligible population, the sampling method, etc., since in this type of study, potential selection bias is practically inevitable and must be evaluated.

Item 2.4. is called methods, which is incoherent since it is a subsection of methods and should have a more specific name. At the same time, citing the primary analyses used might be helpful. Although the programs used are known (SPSS and AMOS), it is always good to indicate the manufacturer.

Although ethical aspects are mentioned, it would be helpful to dedicate a specific section to them at the end of the methods section.

In the results section, it would again be helpful to show the main results from the most general to the most specific. For example, the number of people recruited is usually the first result to show. A systematic approach that goes from the general to the specific allows the reader to follow the text more easily.

In the limitations section, the authors can better explore the inherent limitations of the study. While it is true that they mention potential selection bias, this is important. The characteristics of their population, selection methods, and others should be noted and, if possible, evaluated, so that the reader can better contextualize the external validity of the conclusions.

Author Response

Thanks for the review.

Round 2

Reviewer 1 Report

Comments and Suggestions for Authors

Thank you for addressing the comments in the revised version of the paper.